# A New Insight into the Bactericidal Mechanism of 405 nm Blue Light-Emitting-Diode against Dairy Sourced *Cronobacter sakazakii*

**DOI:** 10.3390/foods10091996

**Published:** 2021-08-25

**Authors:** Shuyan Wu, Pornchanok Subharat, Gale Brightwell

**Affiliations:** 1AgResearch Ltd., Hopkirk Research Institute, Cnr University and Library Road, Massey University, Palmerston North 4442, New Zealand; Kwang.Subharat@agresearch.co.nz (P.S.); Gale.Brightwell@agresearch.co.nz (G.B.); 2New Zealand Food Safety Science and Research Centre, Tennent Drive, Massey University, Palmerston North 4474, New Zealand

**Keywords:** bactericidal mechanism, antimicrobial, visible light, *Cronobacter sakazakii*

## Abstract

(1) Background: Limited evidence exists addressing the action of antimicrobial visible light against *Cronobacter sakazakii*. Here, we investigated the antimicrobial effects of blue-LED (light emitting diode) at 405 nm against two persistent dairy environment sourced strains of *C. sakazakii* (ES191 and AGRFS2961). (2) Methods: Beside of investigating cell survival by counts, the phenotypic characteristics of the strains were compared with a reference strain (BAA894) by evaluating the metabolic rate, cell membrane permeability, and ROS level. (3) Results: The two environment isolates (ES191 and AGRFS2961) were more metabolic active and ES191 showed dramatic permeability change of the outer membrane. Notably, we detected varied impacts of different ROS scavengers (catalase > thiourea > superoxide dismutase) during light application, suggesting that hydrogen peroxide (H_2_O_2)_, the reducing target of catalase, has a key role during blue light inactivation. This finding was further strengthened, following the observation that the combined effect of external H_2_O_2_ (sublethal concentration) and 405 nm LED, achieved an additional 2–4 log CFU reduction for both stationary phase and biofilm cells. (4) Conclusions: H_2_O_2_ could be used in combination with blue light to enhance bactericidal efficacy and form the basis of a new hurdle technology for controlling *C. sakazakii* in dairy processing plants.

## 1. Introduction

*Cronobacter sakazakii* (*C. sakazakii*) is an opportunistic foodborne pathogen isolated from a variety of food sources [1]. Neonates are the most susceptible to fatal infections (e.g., meningitis and septicaemia), although other immune-compromised groups, such as the elderly, are also vulnerable. The organism is a food safety risk in powdered milk products, especially powdered infant formula (PIF) products, as *C. sakazakii* is heat tolerant, resistant to desiccation and can form biofilms, and thus, it may survive in the processing environment for a long time [2]. Environmental isolates of *C. sakazakii* survive via stress response adaptations, making it difficult to be eradicated from a dairy processing environment.

Development of new sanitization technologies is contributing to the improved control of *C. sakazakii*. Osaili et al. reported that 2 kGy of gamma radiation could eliminate 1.2 to 1.5 log CFU of desiccation-stressed *C. sakazakii* isolates in dry infant milk formula [3]. Inactivation of stressed *C. sakazakii* after starvation, heat, cold, acid, alkaline, chlorine or ethanol treatments required 0.82 to 1.24 kGy of gamma radiation with D_10_ value ranging from 1.35 to 1.95 [4]. Ultraviolet (UV 200–260 nm) treatment at 2160 mJ/cm^2^ reduced the bacterial level to the limit of detection (1.18 log CFU per coupon) on silicone, polycarbonate, and stainless steel [5]. Another study reported that in vitro treatment of *C. sakazakii* with conventional UV lamps (254 nm) achieved 1.3log CFU reduction at ambient temperature (25 °C) although higher temperatures (56–57 °C) resulted in higher reductions [6]. However, the safety risks associated with UVC exposure its application as a germicidal technology.

Pulsed broad white light (Xenon flash lamp 190–1100 nm) has been found to achieve 4 log CFU reductions or above of *C. sakazakii* [7,8]. However, it is still a UV-containing application. Visible light (especially blue light wavelengths from 405 to 465 nm) has been shown to have antimicrobial properties and inactivates microbes via photodynamic inactivation, where cellular molecules known as photosensitizers produce reactive oxygen species which react with cellular constituents such as lipids, proteins, and DNA to bring about a cytotoxic effect [9]. Our previous study showed that blue-LED at 405 nm caused a significant reduction (to extinction) in the levels of *Escherichia coli* (Shiga-toxin producing *E. coli*) in vitro [10]. With respect to *C. sakazakii*, there is only limited evidence on its inactivation by blue-LED (light emitting diode) application. A recent study reported that 405-nm LED inactivated *C. sakazakii* in PIF (up to 0.9 log reduction) at a dose of 546 J/cm^2^ and decreased the tolerance of *C. sakazakii* in PIF to desiccation, heat treatment at 50 and 55 °C, low PH (4.75) of simulated gastric fluid, and bile salt [11]. Another study showed that about 375 J/cm^2^ of 405 nm LED illumination significantly reduced the population of a *C. sakazakii* (ATCC 29004) biofilm by 2.0–2.5 log CFU at different temperatures (25, 10, and 4 °C) [12]. In this study, we investigated the antimicrobial effects of blue-LED at 405 nm against three strains of *C. sakazakii*, one reference strain and two isolates from obtained from dairy processing environments. Further, phenotypic characteristics (such as metabolic rate, the outer membrane permeability, and ROS production) when exposed to blue-LED treatment were compared among the tested strains to better understand the antimicrobial mechanism of blue light.

## 2. Materials and Methods

### 2.1. Bacterial Strains

Three *C. sakazakii* strains, ATCC BAA-894 (isolated from powdered formula associated with a neonatal intensive care unit outbreak), ES191 and AGRFS2961 (dairy environment isolates), were used in the study. ES191 has previously been identified as super biofilm former in different nutrient matrices at 37 °C [13]. AGRFS2961 is an isolate from bacterial collection of Food System Integrity Team (AgResearch, Palmerston North, New Zealand) and was isolated from a New Zealand dairy processing plant. Single colonies of each strain were statically incubated under aerobic conditions in tryptic soy broth (TSB) or sterile 10% skim milk at 37 °C for 20 h to obtain overnight bacterial culture (~8 log CFU/mL).

### 2.2. Light Source and Application

Blue light (45 nm) was produced by a prototype LED array, a surface-mounted diode panel consisting of 400 LED bulbs arranged in strips (450 × 450 mm). The cell suspension or cell inoculated coupons were placed vertically at a distance of 25 cm from the light source, which corresponded to average light intensities of 50 mW/cm^2^. Light intensity was measured as described in Wu et al. [10]. The light treatments were all conducted in a temperature-controlled chamber (SKOPE PG600 VC, NZ). The average running temperature and humidity of the lighting surface were recorded per second using customized sensors and software (AgResearch Engineering Development Centre, Lincoln, New Zealand) during light exposure. Surface temperature under illumination was maintained at 28 ± 2 °C; surface humidity was between 20% and 30%.

### 2.3. Cytotoxicity/Antimicrobial Effect of LED against Stationary Phase Cells

The cytotoxicity of blue-LED light was measured using the XTT (sodium 3′-[1-[(phenylamino)-carbony]-3,4-tetrazolium]-bis(4-methoxy-6-nitro)benzene-sulfonic acid hydrate) assay against, tested cells over a 4 h time course [14]. The antimicrobial effect of blue LED illumination was determined by the enumeration of viable bacterial cells on agar plates. Approximately 8 log CFU/mL overnight culture cells in TSB were resuspended in phosphate buffered saline (PBS) and 1.5 mL aliquots for each strain were introduced into 5 wells of a 24 well plate (Nunc™). Subsequently, samples in the 24-well plate were exposed to blue light. As a control, another 24-well plate was tested under the same condition but without light illumination. For both treated and control plates, the cell suspensions were sampled at 0, 1, 2, 3, and 4 h during the test. 

The sampled cells were centrifuged to remove the supernatant and each cell pellet was resuspended in 1.5 mL PBS suspension. For the XTT assay, 500 µL of each sample was taken and mixed with 200 µL of 1 mg/mL XTT solution containing 25 nM PMS (phenazine methosulfate, MilliporeSigma, Burlington, MA, USA) for 3-h dark incubation at 37 °C. The change of colorimetric activity was quantitively measured by optical density at 450 nm with a spectrophotometer (MultiScan Go, Thermo Scientific, Waltham, MA, USA) as the maximum absorbance of the reduced XTT is around 450 nm. A reference reading on cell-free XTT solution with PMS was conducted for correction. Meanwhile, the number of viable cells was enumerated by plate counting (10-fold serial dilution for plating on TSB agar for 24 h incubation at 37 °C) using 1 mL of the cell suspension for each sample.

### 2.4. Change of the Outer Membrane Permeability When Exposed to Antimicrobial Light 

Aliquots of 1 mL were taken from a PBS suspension containing stationary phase cells (7 log CFU/mL) and added into 24-well plates (Nunc^TM^). For each strain, one plate received blue-LED treatment and the other was a control plate held under the same temperature and humidity condition but without any light exposure. Cells were sampled at time 0 min, 15 min, 30 min, 45 min, 1 h, 2 h, 3 h and 4 h. Sampled cells (1 mL) were centrifuged at 10,000× *g* for 3 min and resuspended in 500 µL of 5 mmol/L HEPE [N-(2-Hydroxyethyl) piperazine-N′-(2-ethanesulfonic acid] buffer. One hundred μL of bacterial suspension was mixed with 100 μL of 40 μmol/L NPN (1-N-phenylnapthylamine, Sigma, Burlington, MA, USA) in 96 well plates (Nunc™) for fluorescence measurement at an excitation wavelength of 350 nm and emission wavelength of 420 nm (Varioskan™ LUX, Thermo Scientific, Waltham, MA, USA) while the HEPE buffer mixing with NPN (1:1 ratio) was used as blank.

### 2.5. Endogenous ROS Levels

One ml of a suspension of 7 log CFU/mL stationary phase cells in PBS of each strain was incubated with 2′, 7′-dichlorfluorescein-diacetate, Sigma (DCFH–DA), 2 μM, final concentration for 10 min at 25 °C and followed by a 4-h light application. Samples were collected at 0 min, 15 min, 30 min, 45 min, 1 h, 2 h, 3 h and 4 h and then resuspended in PBS to remove extracellular ROS. A total 200 µL of each sample was transferred into a 96 well plate (Nunc™) and measured for fluorescence intensity using a fluorescence microplate reader (Varioskan™ LUX, Thermo Scientific, Waltham, MA, USA) at an excitation wavelength of 485 nm and emission wavelength of 535 nm [15]. PBS incubated with (2 μM) DCFH–DA was blank as a negative reference.

### 2.6. The Impact of External ROS Scavengers during Light Inactivation

Three different ROS scavengers were investigated, including two microbe-sourced ROS scavenger enzymes, catalase (*Aspergillus niger*, Sigma, Burlington, MA, USA) and superoxide dismutase (*E. coli* SOD, Sigma) and thiourea, (MilliporeSigma, Burlington, MA, USA) [16]. For each strain, 6 log CFU/mL stationary phase cell suspensions containing different final concentrations of ROS scavenger were prepared. One ml aliquots of cell suspensions containing 212/424/848U catalase, 105/525/1050U SOD, and 10/50/100 mM thiourea were placed into each well of 24 well plates and treated with blue-LED illumination for 4 h, at a dose of 720 J/cm^2^. The cytotoxic effect of the applied ROS scavenger agent was also estimated using a replicate plate as control (no light application under the same conditions). A cell suspension containing no external ROS scavenger was also included as a reference group for each strain. All groups were sampled, and serial 10-fold dilutions were prepared for viable cell enumeration. All plates were incubated at 37 °C for 48 h. The bacterial log CFU reduction per ml was determined.

### 2.7. The Combined Effect of Sublethal Hydrogen Peroxide (H_2_O_2_) and Light Inactivation against Stationary Phase Cells from Overnight Culture in 10% Sterile Skim Milk Media 

Suspensions of 8 log CFU/mL cells of each strain were obtained after 20 h overnight incubation at 37 °C in sterile 10% skim milk media (Difco) and resuspended in PBS. One ml aliquots of diluted cell suspension (5 log CFU/mL) containing 0.02% H_2_O_2_ (final concentration) were added into a 24 well plate and subsequently exposed to light treatment for 1 h (180 J/cm^2^). A replicate plate received no light as a control to estimate the effect of H_2_O_2_ itself. Cell suspensions without the addition of H_2_O_2_ or light application were included as a reference. After the experiment, the cell suspension in each well was collected by centrifuging at 10,000× *g* for 5 min and then resuspended in PBS immediately for cell enumeration by plate counting on TSB agar for 48 h incubation at 37 °C. The bacterial log CFU reduction per ml was determined.

### 2.8. Against 3-Day-Old Biofilm Cells Grown from 10% Sterile Skim Milk

Sterile food grade stainless steel coupons (type 304, diameter of 2.5 cm, Agresearch Ltd., Palmerston North, New Zealand) and POM plastic (Sustarin C^®^ polyoxymethylene copolymer commonly used for food conveyors, diameter of 2.5 cm, DOTMAR Ltd., Palmerston North, New Zealand) coupons were placed in 6-well cell culture plates (Nunclon Delta^TM^). Coupon-containing wells were filled with 4 mL of sterile 10% skim milk media. Two hundred µL of 7 log CFU/mL stationary phase cells from an overnight culture in 10% skim milk was added into wells for each strain. Biofilms were grown on stainless steel and plastic coupons for 72 h at 37 °C statically. The 3-day-old biofilm on each coupon was washed in 30 mL of PBS three times to remove curdled milk and unattached bacterial cells. The blue-LED only treated coupons were immediately exposed to light illumination. For LED + H_2_O_2_ groups, external H_2_O_2_ (0.02%) was introduced before the light application by individually dipping each coupon into 0.02% H_2_O_2_ solution (100 mL) for 3 s then transferring into new 6 well plates for light illumination. A reference group was also set up, which consisted of coupons treated with 0.02% H_2_O_2_ in the absence of light illumination. All coupons were sampled by swabbing at 0, 15, 30, 45, and 60 min following light illumination (180 J/cm^2^), and the swabs were re-suspended in 10 mL 0.1% peptone H_2_O plus sterile glass beads. Undiluted sample suspension and 10-fold serial dilutions were plated on tryptic soy agar (TSA) plates and incubated at 37 °C for 48 h for counting detectable colonies.

### 2.9. Statistical Analysis

All experiments were performed in triplicate with biological duplicates. Experimental data were subjected to one-way ANOVA with Tukey’s HSD on Minitab 19 Statistical Software. Statistical significance was determined when *p* < 0.05.

## 3. Results

### 3.1. Cytotoxicity/Antimicrobial Efficacy of LED against Stationary Phase Cells of C. sakazakii

XTT assays were used to examine changes in the metabolic rate of the bacterial populations (stationary phase cells). The number of viable or recoverable cells post treatment were determined in the same samples simultaneously. The light susceptibilities of test strains were different in PBS: BAA894 > AGRFS2961 > ES191. Post 2 h of light treatment, BAA894 showed increased cell loss compared to the other two strains (*p* < 0.05). The total light dose of 720 J/cm^2^ (at 4 h of light illumination) resulted in a 8.88 log CFU/mL reduction of BAA894, 3.47 log CFU/mL reduction of ES191, and 5.69 log CFU/mL reduction of AGRFS2961 strains (Figure 1a–c). 

Strain BAA894 was the most light-susceptible out of the three strains. Post 1 h of treatment, XTT showed a significant reduction in the overall metabolic activity of the whole population (*p* < 0.05). The number of viable cells reduced in a time-dependent manner during 4-h light illumination (Figure 1a). 

There was no significant cell reduction observed for ES191 strain during the first two hours of light treatment (*p* > 0.05) and metabolic activity was not significantly different from that of the control groups up to 3 h of light illumination (*p* > 0.05) (Figure 1b,e), indicating that strain ES191 cells were metabolically active under the blue-LED stress up to 3 h. The number of viable AGRFS2961cells dropped significantly from 1 hr of light illumination (*p* < 0.05) while the colorimetric activities of XTT in LED treated groups were comparable to that of control cells (*p* > 0.05) (Figure 1c,f).

### 3.2. The Permeability Change of Outer Cell Membrane in Light Application 

NPN in cell-free HEPE buffer provided a reference fluorescent intensity at 7.95 ± 0.64 and the fluorescent intensity of NPN in all cell-containing samples showed reading value > 85 units during the time course. The changes of NPN uptake were not significant for BAA894 strain and AGRFS2961 strain during 4 h of the light application (*p* > 0.05) but was observed for ES191 (*p* < 0.05) (Figure 2). LED treated ES191 cells increased NPN uptake from 45 min of light treatment, and the uptake peak was detected at 1 h of light treatment (*p* < 0.05). Following this, NPN uptake decreased after 1 h. In contrast, unilluminated control cells of ES191 started to uptake NPN after 2 h and the NPN uptake increased until the end of the experiment (*p* < 0.05). The fluorescent intensity of NPN for ES191 remained to be the highest among the three tested strains between 45 min and 2 h of the light treatment (*p* < 0.05).

### 3.3. The Change of Endogenous ROS Levels during Light Treatment

Figure 3 shows that there were no differences in the levels of intracellular ROS between the control and LED-treated groups within 1 h of illumination for each strain (*p* > 0.05). Endogenous ROS production increased between 1 h and 2 h of the light treatment and subsequently dropped between 2 h and 4 h of the light treatment. From 2 h to 4 h of the light treatment, the detectable ROS level in LED treated groups was significantly less than that in individual controls for each strain (*p* < 0.05). Strain ES191 had a significantly higher ROS level compared with the other two strains in light-treated groups (*p* < 0.05).

### 3.4. The Effects of External ROS Scavengers during Light Inactivation

We hypothesized that external ROS scavengers could facilitate the removal of endogenous ROS and thereby increase the survival rate of the bacterial population during the light exposure. In the catalase treatment groups, all three tested strains had significantly improved survival rates compared with the corresponding scavenger-free group (*p* < 0.05) (Figure 4). For AGRFS2961 strain, the effect of 848U catalase on cell survival was significantly higher compared with of 212/424U catalase (*p* < 0.05), whilst the impact of different concentrations of catalase remained unchanged in the other two strains (*p* > 0.05). In the thiourea groups, cell survival was improved significantly in strain BAA894 at the three tested concentrations (*p* < 0.05). For the ES191 strain and AGRFS2961 strain, the impact was minimal at 50 mM thiourea compared with the individual scavenger-free group (*p* < 0.05). The impact of SOD was not significant for all three strains (*p* > 0.05).

### 3.5. The Combined Effect of Sublethal Hydrogen Peroxide (H_2_O_2_) and Light Inactivation against Stationary Phase Cells from 10% Skim Milk

The effect of combining external H_2_O_2_ with light inactivation was evaluated (Figure 5). Both 1 h of light treatment (180 J/cm^2^) and 1 h of 0.02% H_2_O_2_ incubation were found to be sublethal to all the tested strains. However, when combined together a significant reduction (>6 log CFU/mL cells during 1-h treatment) was observed (*p* < 0.05). For LED-only treatment trials (long time exposure up to 4 h), ES191 and AGRFS2961 were inactivated with 3 and 4 log CFU/mL reductions respectively at 720 J/cm^2^ whilst a 6 log CFU/mL reduction was observed for BAA894 (Figure 4, control groups).

### 3.6. The Combined Effect of Sublethal Hydrogen Peroxide (H_2_O_2_) and Light Inactivation against 3-Day-Old Biofilm Cells Grown from 10% Skim Milk

Biofilm cells of reference strain BAA894 were not detectable on stainless steel after 3 h light treatment, nor on POM (plastic) after 2 h. The two environment isolates, ES191 and AGRFS2961, were found to persist on either stainless steel or POM coupons during 6-h light exposure (Appendix A). This indicates that the biofilms of strain ES191 and AGRFS2961 are more resistant to light inactivation than that of strain BAA894. The addition of 0.02% H_2_O_2_ accelerated the inactivation efficiency of blue light against the biofilm of ES191 and AGRFS2961; a 5–6 log CFU reduction being achieved in 1 h of light treatment. On the stainless-steel coupon, the impact of combining 0.02% H_2_O_2_ was significant (*p* < 0.05) after 15 min of the light treatment for BAA894 and ES191 strain, but significant reduction of AGRFS2961 was only observed from 45 min (*p* < 0.05) (Figure 6a–c). On the POM coupon, combination with external H_2_O_2_ resulted in significant reduction from 45 min during the light treatment for strain BAA894 and AGRFS2961, and from 15 min for ES191 (*p* < 0.05) (Figure 6d–f). Generally, increased biofilm-cell reduction was achieved by combining LED light with 0.02% H_2_O_2_ within 1 h of the light application, and the biofilm cells of strains, ES191 and AGRFS2961 could be inactivated to an undetectable level on plastic (*p* < 0.05).

## 4. Discussion

Huang et al. recently reported that the treatment of stainless steel using 405 nm LED illumination (375 J/cm^2^ of; 26 mW/cm^2^ for 4 h) reduced the population of *C. sakazakii* in biofilm by 2.0 log CFU at 25 °C, 2.5 log CFU at 10 °C, and 2.0 log CFU at 4 °C [12]. We used a surface mounted diode LED prototype with a stronger light intensity (50 mW/cm^2^). At 2 h of light application (360 J/cm^2^), 3 log CFU reductions were observed for all tested strains on both stainless steel and plastic surfaces (Appendix A). A biofilm developed by BAA894 cells was inactivated further to undetectable levels within 3-h light exposure while biofilms of either ES191 or AGRFS2961 persisted over 6 h. These findings indicate that the efficacy of biofilm removal may be related to strain-specific abilities of cell attachment or affinity to different materials. 

The bacterial response to the applied 405 nm light stress varied among the tested strains. For the BAA894 strain, although at a low metabolic rate after 1 h light treatment (Figure 1d), the light stressed cells remained recoverable until they were inactivated by an intracellular ROS level (triggered with extended light dose) that was high enough to cause irreversible cell death [17]. The XTT activity of strain ES191 and AGRFS2961 showed that the whole population could sustain comparable metabolic rates as the control cells during the light treatment (*p* > 0.05) (Figure 1e,f), suggesting that the light treated populations were still metabolically active (*p* < 0.05). The DCFH–DA assay demonstrated that ROS production increased after 1 h of light treatment (where the cell loss became significant) (Figure 1 and Figure 3), supporting the finding that light induced ROS against many different bacteria was time-dependent [9,10]. After 2 h, blue-LED treated samples had a reduced number of viable cells, which led to the decrease of overall ROS levels in the whole population. The ROS level of the control groups was also increased at 1 h. The higher ROS in control population could be corelated with the higher number of viable cells actively responding to electron transportation and respiration. In other words, ROS level per cell could be varied between the irradiated groups and the control groups. However, the assumption needs to confirm by other methods (e.g., flow cytometry) in the future. A review of literature has not revealed which metabolic responses could be associated with the increase of blue-light resistance. 

Notably, we observed that ES191 and AGRFS2961 produced distinct yellow pigment on TSB plates compared with the colony colour of BAA894. The yellow pigment of *C. sakazakii* strain was previously identified with having carotenogenic nature on molecular/chemical level and BAA894 strain has been reported to carry gene clusters responsible for carotenoid biosynthesis [18,19]. Carotenoid could function as antioxidants to remove intracellular ROS and influence cellular membrane structure/fluidity in the harmful environment [20]. The ability of carotenoid biosynthesis or similar pigmentation pathways [6] should be further investigated for ES191/AGRFS2961 to clarify the possible mechanism of antimicrobial-blue-light resistance. We hypothesize that the contribution of pigment may play varied roles in different strains since our NPN test showed that the permeability of the outer cell membrane was distinct between ES191 and AGRFS2961 (Figure 2). Edward et al. [21] demonstrated that the efficiency of NPN binding to phospholipids was influenced by the redox state of cytochrome o incorporated into phospholipid vesicles in *E. coli.* The addition of H_2_O_2_ resulted in the immediate reoxidation of cytochrome o and loss of the NPN fluorescence increase observed [21]. We hypothesize that blue-LED induced ROS, particular H_2_O_2_ (the focus in our study), possibly inhibited NPN uptake in a strain-dependent manner, as evidenced by constant cell permeability in AGRFS2961/BAA894 strain whereas ES191 had a significant change on the cell permeability, resulting in a differentiated light-susceptible phenotype. Another report found that *E. coli* carrying the cloned cytochrome d terminal oxidase complex (membrane-bound cytochrome) are sensitive to near-UV inactivation [22], suggesting that reoxidation of cytochrome could also be linked to the observed change in permeability of *C. sakazakii* outer membrane when exposed to lethal ROS induced by 405nm blue-LED. However, we admitted that the cell permeability measured by NPN was a phenotypic character for the whole tested cell group. The tested population was assumed to be comprised of viable/damaged/inactivated cells and the ratio of cells in different fitness could be in a dynamic change during the test. Therefore, it is important to determine the cell permeability for viable cells or inactivated cells individually. Again, an investigation by flow cytometry would facilitate to understand at this point.

Photosensitization is an intrinsic process that enhances ROS production in bacteria during photoinactivation and can be influenced by using additional photosensitizers or external ROS scavengers [23]. Our study suggests that the antimicrobial activity of 405 nm blue light was associated with the type of ROS as three different ROS scavengers (catalase, thiourea, and SOD) produced different responses (Figure 4). Catalase showed the most significant effect for population survival under a lethal light dose (720 J/cm^2^) compared with thiourea and SOD (*p* < 0.05). Catalases reduce H_2_O_2_ into H_2_O and O_2_. Effective removal/reduction of H_2_O_2_ by catalase is assumed to be critical for the survival of tested *C. sakazakii* strains when exposed to the blue-LED induced stress. Thiourea is known to reduce OH·, which is a type of ROS derived from H_2_O_2_ and its effect in mitigating antimicrobial light efficacy was found to be significant for BAA894 strain than the other two strains (Figure 4), indicating that BAA894 strain could be more sensitive to a particular ROS stress, e.g., H_2_O_2_ associated species. A supplementary experiment determined that our tested strains had different sensitivities to H_2_O_2_ after a long-time incubation (up to 4 h) at 25 °C and strain BAA894 was significantly more sensitive compared with ES191 and AGRFS2961 (Appendix A). Exogenous H_2_O_2_ could participate in cell damaging and could act as an inhibitor of catalase [24]. Our investigation showed that H_2_O_2_ (either endogenous or exogenous) had a significant role during the 405 nm light inactivation (Figure 4, Figure 5 and Figure 6). Biofilm cells of strains ES191 and AGRFS2961 were resistant to 6 h of the light exposure on both stainless steel and POM coupons (Appendix A). ES191 strain was the most light-resistant strain among three tested strains as it is a known super biofilm former [13] and produced higher amounts of EPS than BAA894/AGRFS2961 (Appendix A). However, the combined effect of 0.02% H_2_O_2_ and 405 nm blue-LED (sublethal light dose at 180 J/cm^2^; sublethal concentration of H_2_O_2_ at 0.02%) could increase the inactivation efficiency against persistent biofilm cells by achieving shortened treatment time (reduced from 6 h to 1 h) and additional 2–4 log reduction (Figure 6). Feuerstein et al. reported that the combination of blue light (450–490 nm) and 0.3 mM H_2_O_2_ yielded 96% growth inhibition of *Streptococcus mutans*, whereas single treatments of blue light orH_2_O_2_ only resulted in a bacterial growth reduction of 3% or 30%, respectively [25]. Therefore, the combined of antimicrobial blue light and H_2_O_2_ could enhance the cytotoxic effect and induce irreversible cell damage rapidly. The current evidence could serve as a scientific guidance for future development/adaption of antimicrobial visible LED technology into dairy processing scenarios.

## 5. Conclusions

This study determined that the two NZ environmentally dairy sourced *C. sakazakii* isolates (ES191/AGRFS2961) tended to be more resistant to blue light at 405nm than the reference isolate (BAA894). The endogenous ROS induced inactivation, by antimicrobial blue light was shown to be time-dependent (Figure 3). Both planktonic cells (stationary phase) and biofilm cells of ES191 and AGRFS2961 were more resistant to a single intervention, i.e., blue-LED or 0.02% H_2_O_2,_ but the combined intervention of 1-h blue-LED and 0.02% H_2_O_2_ achieved inactivation of 5–6 log CFU, with reduction to undetectable level also observed in some cases. Our in-vitro data suggested that when applied alone, antimicrobial blue light could be less efficient or effective against resistant *C. sakazakii* in commercial processing scenarios, but inactivation could be significantly improved in the presence of low levels of H_2_O_2._ Thus, antimicrobial light technology has the potential to be used as a safe and effective approach for cleaning-in-place sanitization especially when applied as a hurdle approach.

## Figures and Tables

**Figure 1 foods-10-01996-f001:**
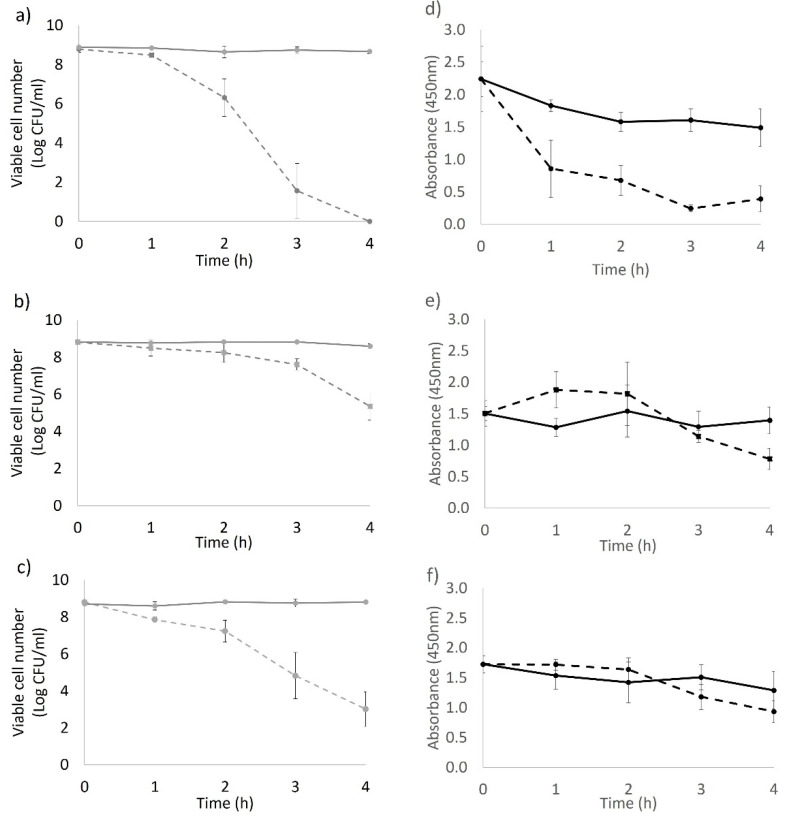
Cytotoxicity/antimicrobial efficacy of 4-h LED application against stationary phase cells of three *C. sakazakii* strains. (**a**) Viable cell counts of BAA894; (**b**) viable cell counts on ES191; (**c**) viable cell counts of AGRFS2961; (**d**) XTT assay on BAA894; (**e**) XTT assay on ES191; (**f**) XTT assay on AGRFS2961; solid line represents control groups and dash line represents LED treated groups.

**Figure 2 foods-10-01996-f002:**
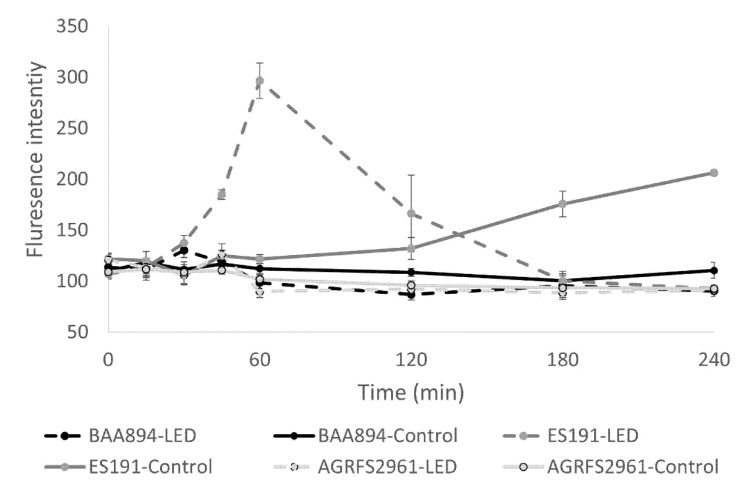
The permeability change of outer cell membrane (NPN uptake) during 4 h of light treatment in vitro.

**Figure 3 foods-10-01996-f003:**
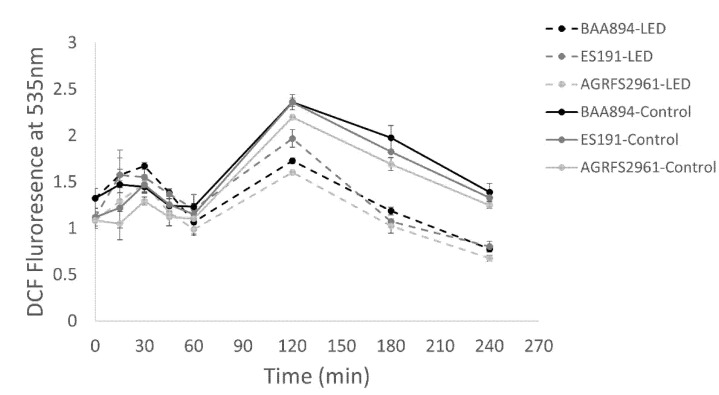
Endogenous ROS detection by DCFH-DA test during 4 h of light application.

**Figure 4 foods-10-01996-f004:**
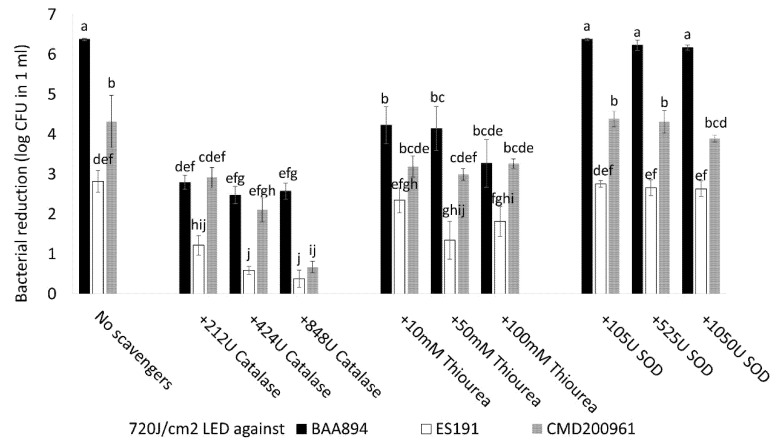
The effects of different ROS scavengers on bacteria log reduction during 720 J/cm^2^ of light treatment. Groups that do not share a letter are significantly different.

**Figure 5 foods-10-01996-f005:**
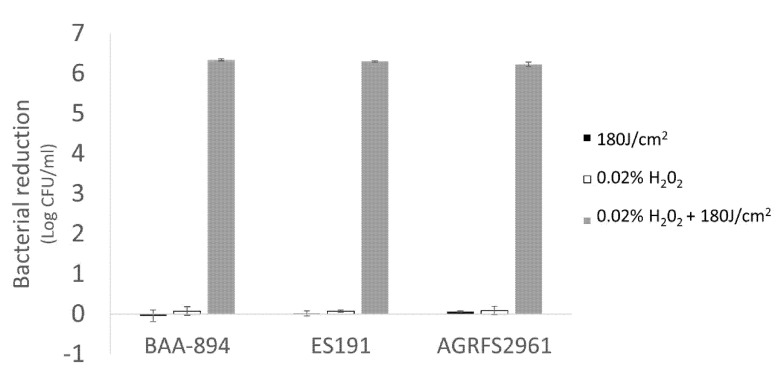
The combined effect of 0.02% H_2_O_2_ and 180 J/cm^2^ of light dose on stationary phase cells.

**Figure 6 foods-10-01996-f006:**
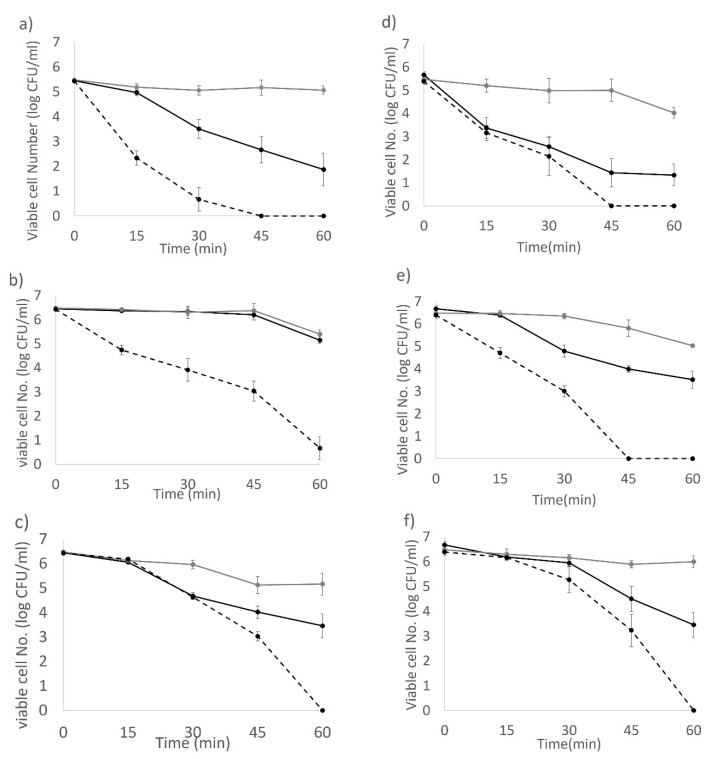
The combined effect of 0.02% H_2_O_2_ and 180 J/cm^2^ of light dose on three-day-old biofilm cells grown from 10% skim milk. (**a**) BAA894 biofilm on stainless steel; (**b**) ES191 biofilm on stainless steel; (**c**) AGRFS2961 biofilm on stainless steel; (**d**) BAA894 biofilm on plastic; (**e**) ES191 biofilm on plastic; (**f**) AGRFS2961 biofilm on plastic; 
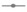
, control groups (with 0.02% H_2_O_2_); 
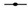
, blue-LED illuminated group; 
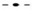
; blue-LED with 0.02% H_2_O_2_.

## Data Availability

The authors choose to exclude this statement as the study did not report any data.

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
