# Peer review of "A New Insight into the Bactericidal Mechanism of 405 nm Blue Light-Emitting-Diode against Dairy Sourced Cronobacter sakazakii"

_foods, 2021, doi:10.3390/foods10091996_

Round 1

Reviewer 1 Report

Cronobacter sakazakii is a known food pathogen and therefore any approach to its reduction is worth investigating. The authors irradiated different C. sakazakii strains with 405 nm in different solutions and in biofilms and determined the illumination results with several techniques. According to the text, a significant CFU reduction is observed with large differences between the strains. Intracelluar ROS concentrations are determined and analyzed and combinations of light and H2O2 examined in milk and biofilms.

Major criticism / questions

  • UnfortunateIy, I cannot read any figure. Contrast and resolution are extremely low.
  • There are many small mistakes, which make the text – at least for me - difficult to read and assess. (They should have been detected by your word processor?)
  • Which is the dominant photosensitizer in C. sakazakii? (Usually porphyrins, but I´m not sure whether C. sakazakii is capable of producing porphyrins?)
  • You observe a permeability change in the cell membrane. Wouldn´t you expect this for dying cells?
  • How do you explain the 1-2h delay in your ROS production? I would have assumed it to be largest in the beginning, and than a reduction due to photobleaching of the photosensitizer? (And how do you explain ROS generation in unirradiated cells. That is very strange?)
  • How do you know the irradiation intensity in milk samples? It should be reduced due to scattering and absorption.

Minor criticism:

  • 45: There is no „low-pressure Ultraviolet“! (There are just Hg-lamps with different pressures but the spectral range you indicate would be fitting to a medium pressure Hg-lamp. But it is seldom used compared to the 254nm low pressure Hg-lamps.)
  • 47: You should introduce “UVC”. (It is not 200-260nm.)
  • 61, 67, 70, … “blue LED” instead of “blue-LED”
  • 88: “405 nm” (the blank between number and unit is missing for half the given values)
  • 91: The light intensity is often an error source, so please describe how you measured the irradiation intensity and please give the reflection properties of your underground, if it is not black! (If it is highly reflective, you might have twice the irradiation you assume.)
  • 96: Were your (unirradiated) reference samples at the same temperature?
  • 115: Why did you measure the optical density at 450 nm? 405 nm is important for your experiment. The difference between both wavelengths might be high!
  • 141: Did you introduce ROS?
  • 292: Explain POM (I would suggest much earlier.)
  • 389: An irradiation with 450 – 490 nm is probably much different to a 405 nm irradiation. At least other photosensitizers are probably involved. (And there is at least another study that claims that there is no synergy if H2O2 and 405 nm are combined.)

Author Response

Major criticism / questions

Comment: UnfortunateIy, I cannot read any figure. Contrast and resolution are extremely low.

Response: The figures have been revised with adjusted font size and resolution. All original pictures of the figures will be attached in a separate Zip file.

Comment:There are many small mistakes, which make the text – at least for me - difficult to read and assess. (They should have been detected by your word processor?)

Response: Proof-reading is finished as requested.

Comment:Which is the dominant photosensitizer in C. sakazakii? (Usually porphyrins, but I´m not sure whether C. sakazakii is capable of producing porphyrins?)

Response: Thanks for the comment. According to the Uniprot database (https://www.uniprot.org/uniprot/A7MJ78), C. sakazakii ATCC BAA-894 possesses hemE gene encoding a uroporphyrinogen carboxylase, which is an enzyme involved in heme biosynthesis.

Comment:You observe a permeability change in the cell membrane. Wouldn´t you expect this for dying cells?

Response: Thanks for the comment. We addressed it in Line 387-391. The fluorescent value represents the change of cell permeability of a population in this experimentation. The tested population was assumed to be comprised of viable/damaged/inactivated cells and the ratio of cells in different fitness could be in a dynamic change during the test. The results showed the cell permeability measured by NPN was a phenotypic character for the whole tested cell group.

As the reviewer suggested, it is important to determine the cell permeability for viable cells or inactivated cells individually. In the future work, an investigation by flow cytometry would facilitate to understand at this point.

Comment:How do you explain the 1-2h delay in your ROS production? I would have assumed it to be largest in the beginning, and than a reduction due to photobleaching of the photosensitizer? (And how do you explain ROS generation in unirradiated cells. That is very strange?)

Response: We agree that prokaryotes are capable of inducing suicide through the free radical (ROS) production and an imbalance between anabolism and catabolism causes significant damage to intracellular components. In our investigation of cell viability, the significant cell loss was not detected until the dose is above 180J/cm2 (Figure 1). The light dose within 180J/cm2 (1 h of light treatment) was considered as sublethal dose for inactivating Cronobacter sakazakii. With dose >180J/cm2 (when the stress became lethal), a burst of excess free radicals generated and lead to cell self-destruction.

We assume that the unirradiated cells had ROS increase because of the growth arrest in the absence of nutrients in the suspension. Other studies reported that any treatment (mild or sublethal stresses) could activate a stress response which is accompanied by free radicals’ production. (Aldsworth, T. G., Sharman, R. L., & Dodd, C. E. R. (1999). Bacterial suicide through stress. Cellular and Molecular Life Sciences CMLS, 56(5), 378-383.)( Cañas-Duarte, S. J., Restrepo, S., & Pedraza, J. M. (2014). Novel protocol for persister cells isolation. PLoS One, 9(2), e88660.)

Free radical accumulation is one of the mechanisms which is associated with cellular respiration. The production of free radicals in the unirradiated cells could be associated with respiration or energy utilization process while the active metabolic response was able to keep the whole population viable during the test. However, the free radical accumulation per cells could be varied between illuminated group and the control group as the number of viable cells in the two groups are different. We addressed it in Line 356-359. (Phillips-Jones, M. K., & Rhodes-Roberts, M. E. (1991). Studies of inhibitors of respiratory electron transport and oxidative phosphorylation. Society for Applied Bacteriology. Technical Series, 27, 203-224.)

Comment:How do you know the irradiation intensity in milk samples? It should be reduced due to scattering and absorption.

Response: We agree that food matrix could absorb partial of the light energy and density of the food matrix also affect the light penetration (Wu, S., Ross, C., Hadi, J., & Brightwell, G. 2021. In vitro inactivation effect of blue light emitting diode (LED) on Shiga-toxin-producing Escherichia coli. Food Control, 125, 107990.), which could be a following work to evaluate for different milk products specifically.

This work focused on in-vitro efficacy of antimicrobial light against two persistent environmental isolates and the potential for surface decontamination. In this study, we did not test the light efficacy in milk samples. The bacterial cells were cultured in sterile milk to obtain stationary phase cells or 3-day-old biofilm cells. The cells were resuspended, or cell-attached-surfaces were washed to remove the milk nutrient before the light application. We addressed it in Line 175-176; Line 194-196.

Minor criticism:

Comment Line 45: There is no „low-pressure Ultraviolet“! (There are just Hg-lamps with different pressures but the spectral range you indicate would be fitting to a medium pressure Hg-lamp. But it is seldom used compared to the 254nm low pressure Hg-lamps.)

Response: We have removed the wrong definition as the reviewer suggested.

Comment Line 47: You should introduce “UVC”. (It is not 200-260nm.)

Response: Thank for the suggestion. The UVC light (200-260 nm) was a light prototype reported in a case study (Jo, S.-H.; Baek, S.-B.; Ha, J.-H.; Ha, S.-D. Maturation and survival of Cronobacter biofilms on silicone, polycarbonate, and stainless steel after UV light and ethanol immersion treatments. Journal of food protection 2010, 73, 952-956.)

This part is to introduce the sanitization technologies in other reported studies which has been used for inactivating the target bacteria. We highlighted the key information for each study.

Comment Line 61, 67, 70, … “blue LED” instead of “blue-LED”

Response: We have made the changes accordingly.

Comment Line 88: “405 nm” (the blank between number and unit is missing for half the given values)

Response: We have made the changes accordingly.

Comment Line 91: The light intensity is often an error source, so please describe how you measured the irradiation intensity and please give the reflection properties of your underground, if it is not black! (If it is highly reflective, you might have twice the irradiation you assume.)

Response: In Line 109-110 We have cited our previous study for using the same light application method and calculating the light doses. To address please see additional test line 111-113 in revised manuscript, we mentioned that the work area in the confined chamber was fully covered with black paper to eliminate light reflection, and the application was always conducted in the darkness. (Wu, S., Ross, C., Hadi, J., & Brightwell, G. 2021. In vitro inactivation effect of blue light emitting diode (LED) on Shiga-toxin-producing Escherichia coli. Food Control, 125, 107990.)

Comment Line 96: Were your (unirradiated) reference samples at the same temperature?

Response: Yes, the control stayed under the same condition during the test. We addressed it in Line 112-113.

Comment Line 115: Why did you measure the optical density at 450 nm? 405 nm is important for your experiment. The difference between both wavelengths might be high!

Response: It is to measure the generation of reduced XTT compounds which has the maximum light absorbance at 450m. We addressed it in Line 122.

Comment Line 141: Did you introduce ROS?

Response: Yes, We addressed it in Line 57-61.

Comment Line 292: Explain POM (I would suggest much earlier.)

Response: The explanation is now in Line 189.

Comment Line 389: An irradiation with 450 – 490 nm is probably much different to a 405 nm irradiation. At least other photosensitizers are probably involved. (And there is at least another study that claims that there is no synergy if H2O2 and 405 nm are combined.)

Response: Regardless of different photosensitizers at wavelengths of 450nm and 490nm, the inactivation will be caused eventually through ROS production. A burst of ROS will be responsible of performing the antimicrobial activity of the light. Therefore, our finding with the light prototype at 405nm was consistent with that in Feuerstein et al.’s study (Reference 27).

Reviewer 2 Report

Comments to the Authors

Manuscript ID: foods-1339188
Title: A new insight into the bactericidal mechanism of 405nm blue light-emitting-diode against dairy sourced Cronobacter sakazakii

Submitted to Foods

I have a number of comments that I believe the authors should address:

Comment #1 (line 89):    

What is the power of each LED bulbs? Did the authors control the light intensity with a powermeter?

Comment #2 (line 101-103):

The XTT assay was used for determining the metabolic rate of each strain after light treatment. How was determined the time of 4h before reading of the ODs? Did the authors perform a kinetic time for determining the optimal time of incubation with XTT before reading the ODs?

Comment #3 (figure 1):

The quality of the figure 1 is very poor.

Comment #4 (figure 2):

The quality of the figure 2 is very poor. How did the authors explain the most important permeability change of outer cell membrane with the strain ES191 after LED application?

Comment #5 (figure 3):

The quality of the figure 3 is very poor. The authors said that the ROS level in LED-treated bacteria is less important than untreated bacteria. But, on the figure 3, the DCF fluorescence is higher for Control groups than LED-treated bacteria. Normally, the DCF fluorescence is closed to an increase of ROS level.

Comment #6 (figure 4):

The quality of the figure 4 is very poor and not interpretable

Comment #7 (figure 6):

The quality of the figure 6 is very poor

Author Response

Response letter

Manuscript ID: foods-1339188
Title: A new insight into the bactericidal mechanism of 405nm blue light-emitting-diode against dairy sourced Cronobacter sakazakii

Submitted to Foods

I have a number of comments that I believe the authors should address:

Comment #1 (line 89):    

What is the power of each LED bulbs? Did the authors control the light intensity with a powermeter?

Response: The surface mounted diode light at 1W was used for prototype fabrication before calibration. The light intensity for the prototype was standardized by unit of mW/cm2 (addressing it in Line 96, reference 10). The light intensity was controlled by changing the illumination distance (the distance between the light source and illuminated samples), for instance 50mW/cm2 at distance of 25 cm, and 23mW/cm2 at distance of 100cm.

Comment #2 (line 101-103):

The XTT assay was used for determining the metabolic rate of each strain after light treatment. How was determined the time of 4h before reading of the ODs? Did the authors perform a kinetic time for determining the optimal time of incubation with XTT before reading the ODs?

Response: For each strain, aliquots of cell suspension were added into 5 wells of two 24 well plates. One plate is for light treatment, and the other is used as control. For both treated and control plates, the cell suspensions were sampled from one well each at 0, 1, 2, 3, and 4 h during the test. We addressed it in Line 111-116. Sampled cells were mixed with XTT/PMS in a new plate and followed with incubation in darkness at 37°C for 3h to render color (addressing it in Line 120).

The optimal time of incubation with XTT/PMS could be a range, for instance usually between 2 and 3 h for our study subject (Cronobacter sakazakii).

Comment #3 (figure 1):

The quality of the figure 1 is very poor.

Response: The figures have been revised with adjusted font size and resolution. All original pictures of the figures will be attached in a separate Zip file.

Comment #4 (figure 2):

The quality of the figure 2 is very poor. How did the authors explain the most important permeability change of outer cell membrane with the strain ES191 after LED application?

Response: The results showed the cell permeability measured by NPN was a phenotypic character for the whole tested cell population. This is the initial evidence that different cell permeabilities were observed on different environmental isolates. Further work will be required to understand the metabolic mechanism in the distinct phenotypes on cellular level rather than on the populational level.

Comment #5 (figure 3):

The quality of the figure 3 is very poor. The authors said that the ROS level in LED-treated bacteria is less important than untreated bacteria. But, on the figure 3, the DCF fluorescence is higher for Control groups than LED-treated bacteria. Normally, the DCF fluorescence is closed to an increase of ROS level.

Response: According to the suggestion, we addressed the point in Line 357-360. In Figure 1, the number of viable cells was significantly reduced after 1 h of treatment in each light-treated group while the whole population of control groups was sustained during the test. The higher ROS in control population could be corelated with the higher number of viable cells actively responding to electron transportation and respiration. In other words, ROS level per cell could be varied between the irradiated groups and the control groups. However, the assumption needs to confirm by other methods (e.g. flow cytometry) in the future.

Comment #6 (figure 4):

The quality of the figure 4 is very poor and not interpretable

Response: The figures have been revised with adjusted font size and resolution. All original pictures of the figures will be attached in a separate Zip file.

Comment #7 (figure 6):

The quality of the figure 6 is very poor

Response: The figures have been revised with adjusted font size and resolution. All original pictures of the figures will be attached in a separate Zip file.

Round 2

Reviewer 2 Report

Manuscript ID: foods-1339188
Title: A new insight into the bactericidal mechanism of 405nm blue light-emitting-diode against dairy sourced Cronobacter sakazakii

Submitted to Foods

The authors have answered to all suggestions or questions.